# Increased IL-17A Serum Levels and Gastric Th17 Cells in *Helicobacter pylori*-Infected Patients with Gastric Premalignant Lesions

**DOI:** 10.3390/cancers15061662

**Published:** 2023-03-08

**Authors:** Chiara Della Bella, Sofia D’Elios, Sara Coletta, Marisa Benagiano, Annalisa Azzurri, Fabio Cianchi, Marina de Bernard, Mario Milco D’Elios

**Affiliations:** 1Department of Molecular and Developmental Medicine, University of Siena, 53100 Siena, Italy; 2Department of Experimental and Clinical Medicine, University of Florence, 50134 Florence, Italy; 3Department of Clinical and Experimental Medicine, University of Pisa, 56126 Pisa, Italy; 4Department of Biology, University of Padua, 35121 Padua, Italy; 5Laboratory of Clinical Pathology, Toscana Centro Hospital, 50100 Florence, Italy

**Keywords:** *Helicobacter pylori*, T cells, cytokines: gastric metaplasia, gastric dysplasia, IL-17

## Abstract

**Simple Summary:**

*Helicobacter pylori* infection represents the major cause of gastric cancer, and is a type I carcinogen for distal gastric cancer. Gastric oncogenesis is a multi-step process and is related at least in part to a peculiar long-lasting gastric inflammation, which is still only partially understood. The aim of this study was to investigate which type of inflammation occurs in the stomach of *Helicobacter pylori* patients with gastric intestinal metaplasia and dysplasia, as well as to examine the serum levels of interleukin 17 in the same patients. We found that *Helicobacter pylori* is able to drive gastric IL-17 inflammation in gastric intestinal metaplasia and dysplasia in *Helicobacter pylori*-infected patients, and that IL-17A serum levels are significantly increased in patients with gastric intestinal metaplasia and dysplasia. We suggest that measurement of serum IL-17A might be useful for the management of *Helicobacter pylori*-infected patients, and eventually for predicting the development of gastric cancer.

**Abstract:**

*Background:* *Helicobacter pylori* infection is characterized by an inflammatory infiltrate that might be an important antecedent of gastric cancer. The purpose of this study was to evaluate whether interleukin (IL)-17 inflammation is elicited by gastric T cells in *Helicobacter pylori* patients with gastric intestinal metaplasia and dysplasia (IM/DYS). We also investigated the serum IL-17A levels in *Helicobacter pylori* patients with gastric intestinal metaplasia and dysplasia, and patients with *Helicobacter pylori* non-atrophic gastritis (NAG). *Methods:* the IL-17 cytokine profile of gastric T cells was investigated in six patients with IM/DYS and *Helicobacter pylori* infection. Serum IL-17A levels were measured in 45 *Helicobacter pylori*-infected IM/DYS patients, 45 *Helicobacter pylori*-infected patients without IM/DYS and in 45 healthy controls (HC). *Results:* gastric T cells from all IM/DYS patients with *Helicobacter pylori* were able to proliferate in response to *Helicobacter pylori* and to produce IL-17A. The Luminex analysis revealed that IL-17A levels were significantly increased in *Helicobacter pylori* IM/DYS patients compared to healthy controls and to *Helicobacter pylori* gastritis patients without IM/DYS (452.34 ± 369.13 pg/mL, 246.82 ± 156.06 pg/mL, 169.26 ± 73.82 pg/mL, respectively; *p* < 0.01, *p* < 0.05). *Conclusions:* the results obtained indicate that *Helicobacter pylori* is able to drive gastric IL-17 inflammation in IM/DYS *Helicobacter pylori*-infected patients, and that IL-17A serum levels are significantly increased in *Helicobacter pylori*-infected patients with IM/DYS.

## 1. Introduction

*Helicobacter pylori* infects more than half of the population worldwide, resulting in lifelong gastric infections, and is the main cause of gastric diseases, such as gastric adenocarcinoma, autoimmune gastritis, lymphoma of gastric mucosa associated lymphoid tissue (MALT), gastric autoimmunity and gastric ulcer [1,2,3,4,5,6,7,8,9,10,11]. *Helicobacter pylori* is classified as a distal gastric cancer carcinogen of type I by the World Health Organization [12]. The incidence of gastric cancer is reduced following eradication of *Helicobacter pylori* [13,14,15,16]. A complex and multifaceted inflammation arises in the stomach due to *Helicobacter pylori* infection, with mononuclear and polymorph nuclear cell infiltration and in situ production of many inflammatory mediators, such as different cytokines and chemokines [17]. *Helicobacter pylori* infection might be faced by the host’s immune system or may escape the host’s defense and eventually lead to gastric cancer [18]. Some *Helicobacter pylori* factors, such as the cytotoxin VacA and the gamma-glutamyl transpeptidase, are even able to immune suppress both innate and acquired immunity by acting both on antigen-presenting cells and T cells [19,20,21,22]. The inflammatory cytokines and chemokines might have several detrimental consequences on gastric epithelial cells, such as unbounded proliferation, evasion of apoptosis, and damage of DNA which eventually leads to malignant transformation [23,24,25].

The carcinogenic pathway of gastric cancer, tightly related to inflammation, is a multistep process that occurs in nearly 1% of patients infected by chronic *Helicobacter pylori* infection. It is of note that while non-atrophic gastritis (NAG) develops in almost all *Helicobacter pylori*-infected patients, in some individuals, the inflammation advances further to chronic atrophic gastritis stages of pre-malignancy, such as gastric intestinal metaplasia (IM) and gastric dysplasia (DYS) [26,27]. Different types of genetic instabilities, characterized by multiplex and dangerous clonal composition, represent the latter steps of a gastric inflammatory-related carcinogenic process [28,29,30,31,32].

It has been recently shown that there is a predominant T helper 17 response in *Helicobacter pylori*-infected patients with gastric adenocarcinoma and in *Helicobacter pylori* negative patients with autoimmune atrophic gastritis [33,34,35,36,37]. However, there are no clues on the pathogenic process mediated by *Helicobacter pylori* specific T helper 17 lymphocytes in human gastric intestinal metaplasia and gastric dysplasia. Inflammatory mediators (such as cytokines) which in situ originate in the stomach may diffuse into the blood and then be detected [38]. Serum cytokine levels of gastric premalignant patients somehow mirror the inflammatory *milieu*, which is ongoing in the gastric mucosa, suggesting that measurement of serum IL-17A might be useful for monitoring the mucosa inflammation of *Helicobacter pylori*-infected individuals. We thus aim to investigate the in vivo production of IL-17A by gastric T cells obtained from *Helicobacter pylori* patients with gastric intestinal metaplasia and gastric dysplasia. As we wondered whether IL-17A could be abnormal in patients with IM/DYS, we further measured serum IL-17A in *Helicobacter pylori* patients with gastric intestinal metaplasia and gastric dysplasia, in non-atrophic gastritis *Helicobacter pylori* patients without IM/DYS and in healthy controls.

## 2. Materials and Methods

### 2.1. Patients

Upon approval from the local Ethical Committee (ethic statement n.14936/CAM_BIO), we investigated the T helper 17 immune responses both at gastric and at serum level in *Helicobacter pylori* patients with gastric intestinal metaplasia and dysplasia (IM/DYS), diagnosed by histology, according to the updated Sydney Houston criteria [39,40]. The intestinal metaplasia was defined by the presence of goblet cells or a brush border. The diagnosis of dysplasia was based on the increased cell proliferation, accompanied by abnormalities in cell size and configuration, such as reduction of mucus secretion, increased nuclear–cytoplasmic ratio, loss of nuclear polarity, and pseudo stratification and architectural derangement of the glands resulting in intraluminal in folding, cellular crowding and glandular branching. The dysplasia was distinguished from the regenerative hyperplasia by non-considering the dysplastic histological changes occurring in the areas of mucosal injury, such as intense gastritis and ulcers [41]. In six patients with IM/DYS and *Helicobacter pylori* infection (three males, three females, mean age 54; range 50 up to 58 years), following informed consent, biopsy specimens were obtained from the gastric mucosa in order to study mucosal T cells. In these six patients with IM/DYS and *Helicobacter pylori* infection and in the other 129 patients or controls, we studied serum levels of IL-17A. Of the 135 subjects, 45 were *Helicobacter pylori* patients with IM/DYS (diagnosed by histology), 45 were *Helicobacter pylori* patients with non-atrophic gastritis (NAG) (diagnosed by histology), and 45 were HC. IM/DYS patients were 24 (53%) males and 21 (47%) females, mean age 67.4 ± 12.3. The NAG patient group was composed of 23 (51%) males and 22 (49%) females and with mean age 68.1 ± 15.8 years. Healthy subjects were 22 (49%) males and 23 (51%) females and with a mean age of 62.6 ± 11.3 years. HC had no gastric or dyspeptic symptoms; they were all *Helicobacter pylori* negative; they did not suffer from peptic ulcer, gastric neoplasia, gastric autoimmunity; and they had no history of gastric medication in the previous two years. All IM/DYS, NAG patients and healthy controls were investigated by serology (Helori CTX, Eurospital, Trieste, Italy). No previous *Helicobacter pylori* eradication therapy had been provided to these patients before they were enrolled in this study. IM/DYS, NAG patients were seropositive for *Helicobacter pylori*, whereas all HC were not. Demographics, clinical characteristics and family history of the study population are detailed in Table 1.

### 2.2. IL-17 Production by Gastric Mucosa T Cells

The IL-17 production by gastric T cells of IM/DYS patients was investigated by both ELISA and ELISpot analysis. Fresh gastric T cells (1 × 10^5^) of each IM/DYS patient were collected and activated by anti CD3, OKT3 clone, immobilized mAb (5 μg/mL) (BioLegend, San Diego, CA, USA) for 72 h of incubation at 37 °C, 5% CO_2_ in humidified atmosphere [33]. Culture supernatants were harvested, and fresh gastric T cells were tested for IL-17 and IL-4 production by ELISA, according to the manufacturer’s instructions (BioSource, San Diego, CA, USA).

T cells from the gastric mucosa of each IM/DYS patient were cultured by adding IL-2 (30 U/mL) every 3 days. On day 10, gastric T cells were tested in triplicate for their IL-17 production by ELISpot assay (eBioscience—Thermo Fisher Scientific, Waltham, MA, USA). In particular, 2.5 × 10^5^ cells were stimulated with medium alone, or *Helicobacter pylori* (NCTCI 1637 strain), or tetanus toxoid (TT) in the presence of autologous antigen presenting cells (APCs). After 48 h of incubation at 37 °C, 5% CO_2_ in humidified atmosphere, the developed spot forming cells (SFCs) were counted with an automated ELISpot reader (AID AutoImmun Diagnostika, Strassberg, Germany) and used as a quantitative measure for the number of IL-17 cytokine secreting cells as described [42].

### 2.3. Characterization of Antigen Specificity and IL-17 Production by Gastric T Cell Clones Obtained from *Helicobacter pylori* Patients with Gastric Intestinal Metaplasia, and Gastric Dysplasia

Gastric mucosal surgical specimens collected from each IM/DYS patient were cultured for 7 days in RPMI 1640 complete medium (BioConcept AG, Allschwil, Swiss) supplemented with L-glutamine 1%, beta-mercaptoethanol 1%, non-essential amino acids 1%, Na-pyruvate 1%, penicillin 50.000 U and streptomycin 50 mg; with addition of 20% of HB basal medium (Irvine Scientific, Santa Ana, CA, USA), 10% of fetal bovine serum (HyClone Laboratories, South Logan, UT, USA), 3% of human serum (Sigma Aldrich, St. Louis, MO, USA) and human recombinant IL-2 (PeproTech, London, UK) 50 U/mL in order to select and amplify the activated T cells infiltrating the biopsy.

Single T cell blasts were cloned under the limiting dilution method, finally seeding a mean number of 0.3 cells in each 96 plate well in the presence of feeder cells (2 × 10^5^ irradiated PBMC), the non-specific T cell receptor mitogen PHA (0.5% *v*/*v*) and human recombinant IL-2 (50 U/mL) [43]. Flow cytometry on BD FACS Canto II with the FACSDiva software (Becton Dickinson, Franklin Lakes, NJ, USA) was performed for the cell surface CD3 subpopulation markers analysis of T cell clones. For this purpose, fluorochrome conjugated anti CD4 PE and anti CD8 FITC antibodies were used for labelling 2 × 10^5^ cells of each T cell clone obtained and for the detection of its helper or cytotoxic surface phenotype.

T cell blasts (5 × l0^4^) from each of the CD4^+^ and CD8^+^ clones were screened in triplicate cultures for their responsiveness to *Helicobacter pylori* by measuring [^3^H] thymidine (Perkin Elmer, Waltham, MA, USA) uptake after 60 h stimulation with medium alone, or with *Helicobacter pylori* antigens (10 μg/mL), or with the non-specific T cell receptor mitogen PHA (1% *v*/*v*) as a positive control [44]. The proliferative responses of each T cell clone for *Helicobacter pylori* were then measured as beta emission due to tritium incorporation, by counts per minute (CPM). The mitogenic index (MI) was determined as the ratio between mean values of CPM obtained in stimulated cultures and those obtained in unstimulated ones (medium alone). Mitogenic index >5 was considered as positive [45].

In order to define the cytokine profile of each *Helicobacter pylori*-specific T cell clone, 5 × 10^5^ T cell blasts were stimulated for 48 h with medium alone or with *Helicobacter pylori* (10 μg/mL) with 5 × 10^5^ irradiated autologous APCs as previously described [46]. The quantitative determinations of IL-17, IFN-γ were performed by ELISA, following the manufacturer’s specifications (BioSource, San Diego, CA, USA).

### 2.4. Luminex Assay for IL-17A

All sera were investigated for IL-17A by Luminex xMAP technology with Bio-Plex Pro™ kit (BIO-RAD, Hercules, CA, USA) and the assay procedure was performed in accordance with the manufacturer’s instructions. Data were detected using the fluorescent imager MAGPIX System (Luminex, Austin, TX, USA) and acquired with the Bio-Plex Manager™ software (Hercules, CA, USA). IL-17A concentration was determined by fitting fluorescence intensity values into the calibration curve; a four-parameter logistic algorithm was used for the best curve fit. IL-17A detection range was LLOQ (lower limit of quantification)–ULOQ (upper limit of quantification) = 1.20–19,682.00 pg/mL.

### 2.5. Statistical Analyses

Using IBM^®^SPSS Statistic (version 28.0, Armonk, NY, USA), descriptive and inferential statistics were performed to summarize and analyze data, respectively.

The Shapiro–Wilk test was applied to assess the normality of IL-17 concentration distributions. Statistical significance among different groups was calculated by Mann–Whitney U test. *p* < 0.05 was considered statistically significant.

Sera IL-17 concentrations falling below the LLOQ of the Luminex test were replaced with one-half the respective LLOQ for descriptive purposes [47]. No subject had values above the ULOQ.

In order to understand the accuracy value of the IL-17A Luminex assay, the graphical ROC (Receiving Operating Characteristic) curve was plotted by matching sensitivity against (1–specificity) for the various values tabulated in each distribution. The number of necessary samples to meet the desired statistical constraints for IL-17A Luminex assay was computed in relation to the number of IM/DYS cases. The Areas Under the ROC Curves (AUC), representing the test discrimination power, were measured [48].

## 3. Results

### 3.1. IL-17A Secretion by Gastric T Cells from *Helicobacter pylori* Patients with Gastric Intestinal Metaplasia and Dysplasia (IM/DYS)

We examined whether gastric T cells from *Helicobacter pylori* patients with IM/DYS produced IL-17 or IL-4. Gastric-derived T cells were analyzed for their IL-17 production both in culture supernatants and ELISpot. After 72 h stimulation with immobilized anti-CD3 mAb, gastric T cells produced a high level of IL-17A (104,163 ± 162.26 pg/mL), whereas there was almost no production of IL-4 (23.91 ± 6.29 pg/mL) (Figure 1A). Gastric T cells from each *Helicobacter pylori* IM/DYS patient were stimulated for 48 h with *Helicobacter pylori* and tested by IL-17 ELISpot; at the end of the culture period, the SFCs count highlighted that a significant amount of T helper cells derived from the gastric mucosa of *Helicobacter pylori* patients with IM/DYS patients produced IL-17A if stimulated with *Helicobacter pylori* antigen compared to the unstimulated ones. No significant concentration of IL-17 was detected in tetanus toxoid stimulated gastric T cells compared to the ones incubated with medium alone for each of the six patients with gastric intestinal metaplasia or dysplasia (Figure 1B).

### 3.2. Predominant T Helper 17 Phenotype of Helicobacter pylori-Specific CD4^+^ T Cell Clones Obtained from *Helicobacter pylori* Patients with Gastric Intestinal Metaplasia, and Gastric Dysplasia

To characterize at clonal level the in vivo activated T cells present in the gastric inflammatory infiltrates of *Helicobacter pylori* IM/DYS patients, T cell blasts were recovered and cloned by limiting dilution. From the stomach of IM/DYS, 242 T cell clones were derived, and they were all characterized for their surface phenotype, antigen specificity and cytokine production. A total of 167 clones out of 242 were CD4^+^ and 75 clones were CD8^+^. *Helicobacter pylori* specificity of T cell clones was investigated by testing their ability to proliferate in response to *Helicobacter pylori* under MHC-restricted conditions. While all 75 CD8^+^ T cell clones were activated by the PHA (positive control) and proliferated to the mitogen, none of them were responsive to the *Helicobacter pylori* (Figure 2A,B). A total of 29 CD4^+^ T cell clones derived from the gastric mucosa of *Helicobacter pylori* IM/DYS showed significant proliferation (mitogenic index > 5) in response to *Helicobacter pylori* (Figure 2C), whereas all CD4^+^ T cell clones proliferated to PHA (Figure 2D). In order to define the cytokine profile of the 29 *Helicobacter pylori*-specific T helper clones following antigen stimulation, IFN-γ and IL-17 production was measured. The total number of T helper *Helicobacter pylori*-specific clones secreted IL-17 in response to *Helicobacter pylori* stimulus, and 11/29 gastric clones produced both IFN-γ and IL-17 (Figure 3).

### 3.3. Serum IL-17A Levels Are Elevated in Sera of *Helicobacter pylori* Patients with Gastric Intestinal Metaplasia, and Gastric Dysplasia

Serum IL-17A levels were quantified by Luminex assay in 45 *Helicobacter pylori* IM/DYS patients, in 45 *Helicobacter pylori* patients without IM/DYS and in 45 healthy controls (HC). The Luminex analysis revealed that IL-17A levels were significantly increased in *Helicobacter pylori* IM/DYS patients compared to healthy controls and to *Helicobacter pylori* gastritis patients without IM/DYS (452.34 ± 369.13 pg/mL, 246.82 ± 156.06 pg/mL, 169.26 ± 73.82 pg/mL, respectively; *p* < 0.01, *p* < 0.05) (Figure 4).

The receiver operating characteristics (ROC) analysis was used to assess the performance of IL-17A Luminex assay in discriminating between *Helicobacter pylori* IM/DYS patients and healthy subjects, and between *Helicobacter pylori* IM/DYS patients and *Helicobacter pylori* patients without IM/DYS. The ROC curves for IL-17A are depicted in Figure 5. The AUC of IM/DYS vs. NAG was 0.62; the AUC of IM/DYS vs. HC was 0.67; and the AUC of NAG vs. HC was 0.64.

## 4. Discussion

*Helicobacter pylori* infection is invariably characterized by gastric inflammation which may lead to gastric intestinal metaplasia, dysplasia and gastric cancer. The present investigation indicates several important findings that are relevant to gastric intestinal metaplasia and dysplasia potentially leading to gastric cancer associated with *Helicobacter pylori* infection: (a) *Helicobacter pylori* promotes gastric T helper 17 inflammatory immune pathological responses in *Helicobacter pylori*-infected patients with gastric intestinal metaplasia and dysplasia; and (b) the serum levels of interleukin 17A are significantly increased in *Helicobacter pylori* patients with gastric intestinal metaplasia and dysplasia compared to *Helicobacter pylori* patients with non-atrophic gastritis, as well as compared to healthy controls.

The type of chronic cytokine secretion going on in the stomach is able to influence the progression towards gastric cancer [25,49,50,51,52,53]. *Helicobacter pylori* is able to induce an innate immune response rich in cytokines able to drive a T cell response towards Th17 phenotype, such as IL-1β, IL-6, transforming growth factor (TGF)-β and IL-23 [54,55]. Th17 inflammatory responses are driven by T helper 17 cells that secrete IL-17, and are key players in the induction of many inflammatory pathways at the gastric level [56]. Among gastric tissues of patients with *Helicobacter pylori* infection, the upregulation of IL-17 has been found [56,57,58]. Several reports demonstrated the relevance of Th17 in gastric adenocarcinoma. It has been demonstrated that *Helicobacter pylori*-specific T lymphocytes produce huge amount of IL-17 in patients with gastric adenocarcinoma [33,35,36]. However, it is still not clear whether *Helicobacter pylori* antigens might be able to drive Th17 responses in *Helicobacter pylori* patients with preneoplastic gastric malignancies, such as gastric intestinal metaplasia and dysplasia. Important findings of the present investigation are that *Helicobacter pylori* IM/DYS patients’ T helper cells specific for *Helicobacter pylori* drive a peculiar gastric inflammation, rich in interleukin 17A. These results prompted us to examine the interleukin-17A amount in the sera of patients with gastric premalignant lesions, gastric intestinal metaplasia and dysplasia, compared to patients with non-atrophic gastritis. A significant increase in serum IL-17A was found in the peripheral blood of subjects with gastric intestinal metaplasia and dysplasia, compared to patients with non-atrophic gastritis and to controls. Thus, IL-17A is not only a key player of gastric mucosa inflammation in patients with autoimmune atrophic gastritis with no *Helicobacter pylori* infection [35], but also a relevant cytokine for the immune pathogenesis of gastric intestinal metaplasia and gastric dysplasia in *Helicobacter pylori*-infected patients on the basis of the findings obtained in the present study, both in the stomach and in the sera. Given that this investigation represents a pilot single-center study, future multi-center research activities dealing with serum IL-17A levels in gastric adenocarcinoma would be very important for a better definition of the test. Gastric intestinal metaplasia and gastric dysplasia are considered precursor lesions of gastric cancer, suggesting an important link between gastric Th17 inflammation and gastric cancer. A large body of results suggests that gastric cancer, due to *Helicobacter pylori*, is strongly related to a multi-step process rich in T helper 17 inflammatory responses that might progress, if unabated, from gastric intestinal metaplasia and dysplasia to gastric cancer [53,59,60,61,62,63].

## 5. Conclusions

The results obtained indicate that *Helicobacter pylori* is able to drive gastric IL-17 inflammation in IM/DYS *Helicobacter pylori*-infected patients, and that IL-17A serum levels are significantly increased in *Helicobacter pylori*-infected patients with IM/DYS.

We propose that the onset of gastric cancer is preceded by decades of gastric T helper 17 inflammation due to *Helicobacter pylori* infection. On the basis of the results obtained so far, we suggest that serum determination of interleukin-17A would be helpful in the clinical practice of managing *Helicobacter pylori*-infected patients, and eventually for predicting the development of gastric cancer.

## Figures and Tables

**Figure 1 cancers-15-01662-f001:**
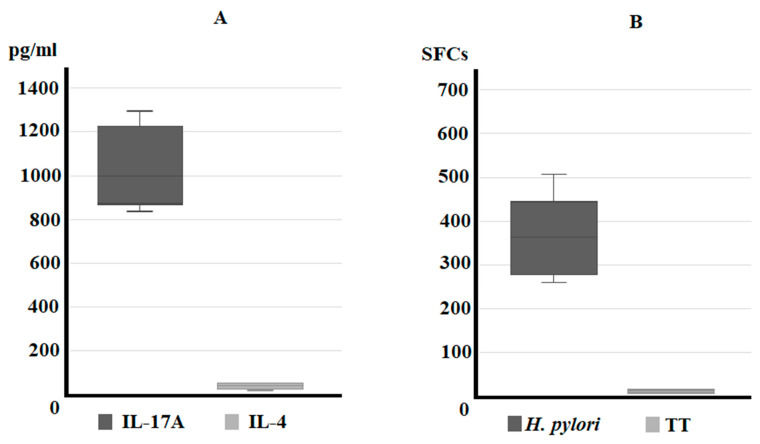
Profile of IL-17 production of IM/DYS *Helicobacter pylori* gastric T cells. ELISA assay for IL-17 and IL-4 secretion (pg/mL) from fresh gastric T cells after 72 h of stimulation with immobilized OKT3 anti-CD3 mAb (**A**). ELISpot assay for IL-17 production (spot forming cells—SFCs for 10^6^ cells) by gastric T cells stimulated with *Helicobacter pylori* or tetanus toxoid (TT) (**B**).

**Figure 2 cancers-15-01662-f002:**
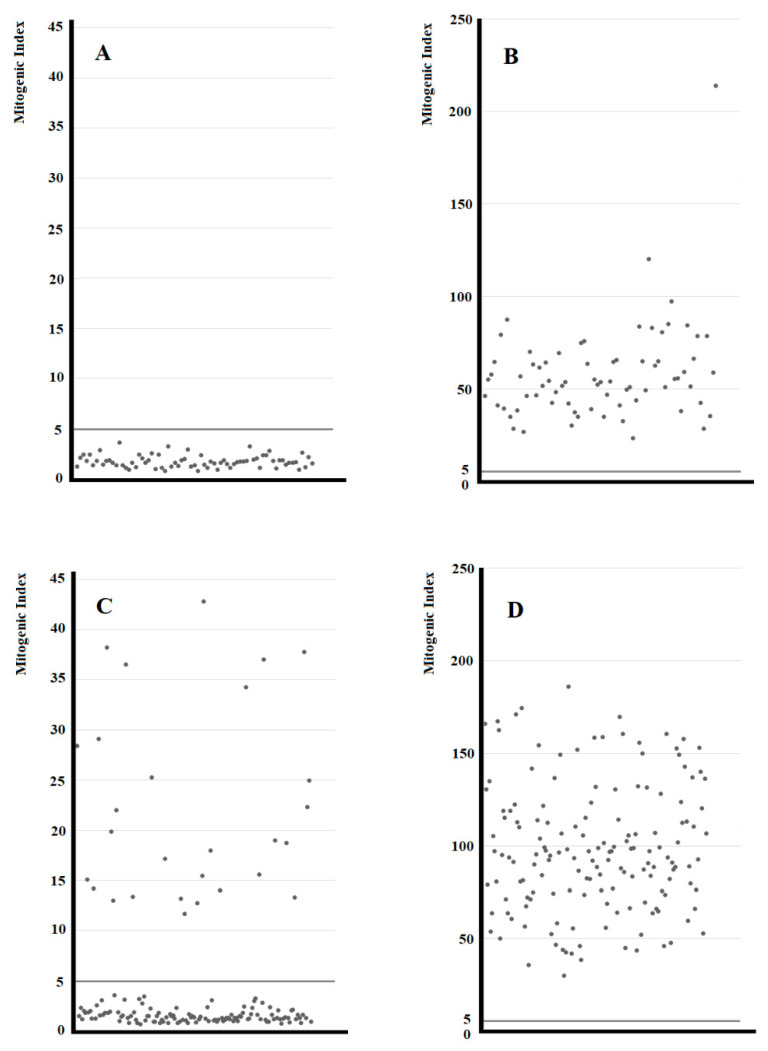
Proliferative response to *Helicobacter pylori* and PHA of the CD8^+^ and CD4^+^ T cell clones from *Helicobacter pylori* patients with gastric intestinal metaplasia and dysplasia (IM/DYS). In vivo-activated T cells were recovered from biopsy specimens of gastric mucosa of patients with IM/DYS. Gastric T cell clones were obtained and tested for their *Helicobacter pylori* specificity and proliferative response: after 60 h co-cultured with irradiated autologous APCs in the presence of medium alone, PHA or the *Helicobacter pylori* NCTCI 1637 strain, the [^3^H]thymidine uptake was measured and expressed as mitogenic index. Proliferative response to *Helicobacter pylori* of the CD8^+^ T cell clones (**A**). Proliferative response to PHA of the CD8^+^ T cell clones (**B**). Proliferative response to *Helicobacter pylori* of the CD4^+^ T cell clones (**C**). Proliferative response to PHA of the CD4^+^ T cell clones (**D**).

**Figure 3 cancers-15-01662-f003:**
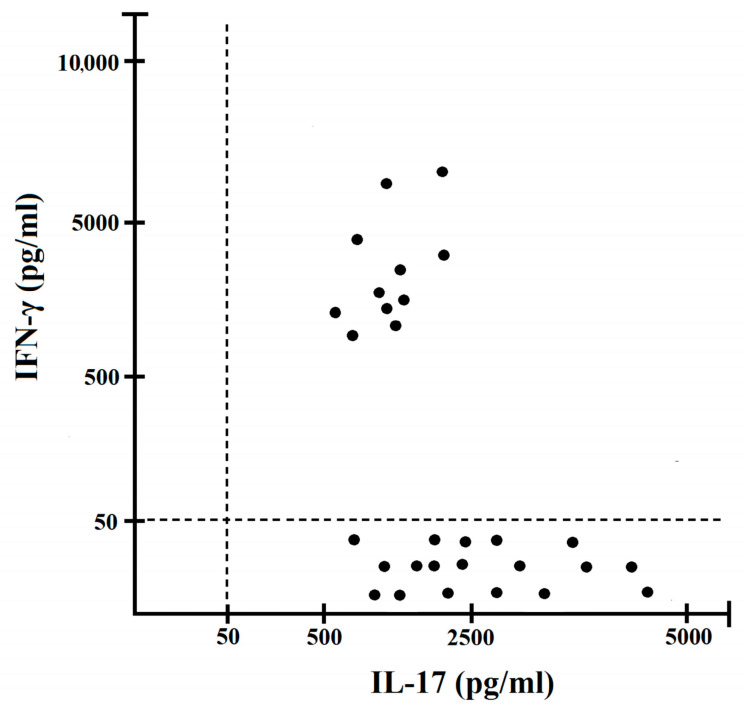
IL-17 and IFN-γ production at clonal level by *Helicobacter pylori*-specific T helper cells derived from the gastric mucosa biopsies of *Helicobacter pylori* IM/DYS patients. Cytokines were measured by ELISA in T cells culture supernatants following *Helicobacter pylori* stimulation. Levels of IL-17, IFN-γ were consistently <20 pg/mL in not stimulated cultures.

**Figure 4 cancers-15-01662-f004:**
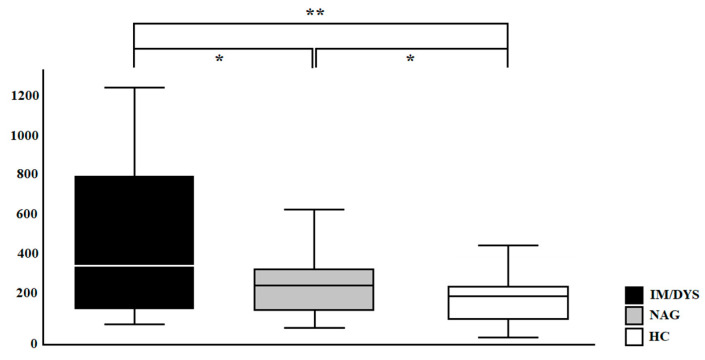
IL-17A (pg/mL) levels in sera samples of enrolled subjects (IM/DYS, *Helicobacter pylori* patients with gastric intestinal metaplasia and dysplasia; NAG, *Helicobacter pylori* patients with non-atrophic gastritis; HC, healthy control). * *p* < 0.05, ** *p* < 0.01.

**Figure 5 cancers-15-01662-f005:**
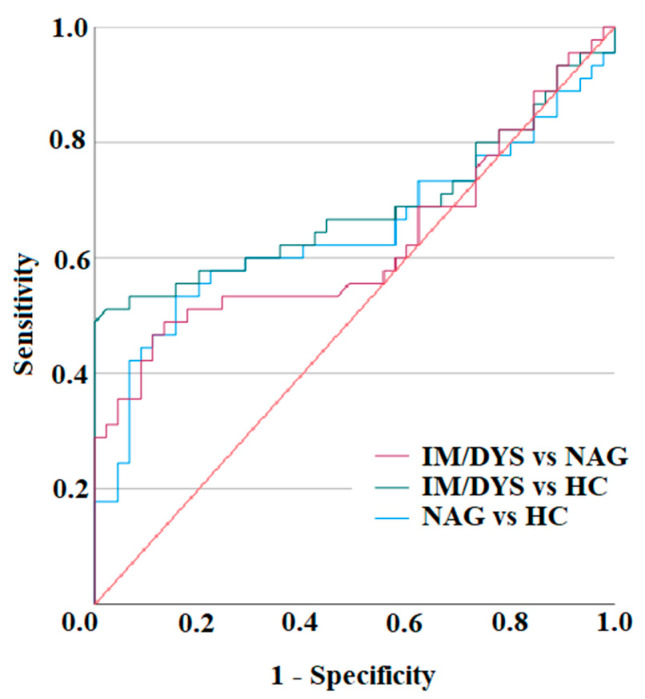
ROC curves of IL-17A Luminex assay. Distributions of the serum amounts of IL-17 were computed by ROC analysis for 45 *Helicobacter pylori* gastric intestinal metaplasia and dysplasia (IM/DYS) patients, 45 *Helicobacter pylori* develop non-atrophic gastritis (NAG) patients, and 45 healthy subjects (HC) to assess the test accuracy.

**Table 1 cancers-15-01662-t001:** Characteristics of patients and controls. IM/DYS gastric intestinal metaplasia and dysplasia, NAG H. pylori non-atrophic gastritis, HC healthy controls, BMI body mass index, GC gastric cancer.

		IM/DYS (N = 45)	NAG (N = 45)	HC (N = 45)
**Age**	<60	20 (44%)	21 (47%)	24 (53%)
	≥60	25 (56%)	24 (53%)	21 (47%)
**Sex**	Female	21 (47%)	22 (49%)	23 (51%)
	Male	24 (53%)	23 (51%)	22 (49%)
**Ethnicity**	Caucasian	45 (100%)	45 (100%)	45 (100%)
**BMI** (kg/m^2^) mean ± SD		23.64 ± 2.61	23.67 ± 2.89	23.69 ± 2.77
**Smoking habits**	Never	34 (75%)	35 (77%)	38 (84%)
	Ever	11 (25%)	11 (23%)	7 (16%)
**Alcohol use**	Never/rare drinker	31 (68%)	32 (71%)	37 (81%)
	Drinker	14 (32%)	13 (29%)	8 (19%)
**Family history of GC**	No	42 (93%)	43 (95%)	45 (100%)
	Yes	3 (7%)	2 (5%)	0 (0%)
**Pernicious anemia**	No	45 (100%)	45 (100%)	45 (100%)
	Yes	0 (0%)	0 (0%)	0 (0%)

## Data Availability

Data supporting reported results are available from the coordinator of the study, Mario Milco D’Elios, corresponding author.

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
