# Peer review of "Increased IL-17A Serum Levels and Gastric Th17 Cells in Helicobacter pylori-Infected Patients with Gastric Premalignant Lesions"

_cancers, 2023, doi:10.3390/cancers15061662_

Round 1

Reviewer 1 Report

The authors of this work study the differential production of IL-17 by T cells of the gastric mucosa of patients infected by H. pylori with and without gastric intestinal metaplasia and dysplasia (IM/DYS), in comparison with healthy controls. They analyze the capacity of the determination of IL-17 in serum as a potential minimally invasive marker to identify the presence of IM/DYS.

The work is, in general, well done and provides a plausible hypothesis for the involvement of Th17 lymphocyte stimulation in precursor lesions of H. pylori-related distal gastric cancer.

Some proposals to improve the presentation of the results and better understand them are proposed to the authors.

1. H. pylori is an important risk factor for the development of gastric cancer, but not the only one. Gastric atrophy with intestinal metaplasia may be due to H. pylori, but also to an autoimmune process (production of anti-gastric parietal cell antibodies), and other risk factors for gastric cancer have been described and have been well characterized. It would be important for the authors to present the characteristics of their study population with respect to risk factors related to gastric cancer, in order to provide evidence of comparable study groups in terms of additional risk factors. To do this, I would recommend adding Table 1 to the manuscript detailing the basic demographic and clinical characteristics of the study population (the 3 groups of patients analyzed), including:

Age, sex, ethnicity, body mass index, tobacco and alcohol use, family history of cancer, other forms of gastropathy (such as pernicious anemia or Menetrier's disease), if possible.

2. H pylori infection status were defined by serology. Please indicate that no previous H. pylori eradication therapy had been provided to these patients before they were enrolled in this study.

3. Likewise, the healthy control subjects recruited for this study must be clearly defined; in them, the absence not only of H. pylori infection (negative serology) but also of gastric mucosal lesions should ideally be excluded.

Author Response

English language and style are fine/minor spell check required
English language has been carefully revised by a professional English translator and all the text has been edited accordingly. Thus, Acknowledgements have been added in the manuscript.

  1. H. pylori is an important risk factor for the development of gastric cancer, but not the only one. Gastric atrophy with intestinal metaplasia may be due to H. pylori, but also to an autoimmune process (production of anti-gastric parietal cell antibodies), and other risk factors for gastric cancer have been described and have been well characterized. It would be important for the authors to present the characteristics of their study population with respect to risk factors related to gastric cancer, in order to provide evidence of comparable study groups in terms of additional risk factors. To do this, I would recommend adding Table 1 to the manuscript detailing the basic demographic and clinical characteristics of the study population (the 3 groups of patients analyzed), including: Age, sex, ethnicity, body mass index, tobacco and alcohol use, family history of cancer, other forms of gastropathy (such as pernicious anemia or Menetrier's disease), if possible.

Table 1 with study population characteristics has been added, as recommended (line 115). As shown in Table 1, the study groups investigated are comparable.

  1. H pylori infection status were defined by serology. Please indicate that no previous H. pylori eradication therapy had been provided to these patients before they were enrolled in this study.

The text “No previous Helicobacter pylori eradication therapy had been provided to these patients before they were enrolled in this study.” (line 110) has been added.

  1. Likewise, the healthy control subjects recruited for this study must be clearly defined; in them, the absence not only of H. pylori infection (negative serology) but also of gastric mucosal lesions should ideally be excluded.

Healthy controls were carefully evaluated from a clinical point of view and they had no gastric or dyspeptic symptoms, they were all Helicobacter pylori negative, they did not suffer from peptic ulcer, gastric neoplasia, gastric autoimmunity and they had no history of gastric medication in the previous two years. These informations have been added in the revised version of the manuscript at line 106.

Reviewer 2 Report

Dear All,  

I was pleased to review the article “ Increased IL-17A Serum Levels and Gastric Th17 Cells in Helicobacter pylori-Infected Patients With Gastric Premalignant Lesions”

This is a well-designed and a well-written article concerning the gastric Th 17 cells and serum levels of IL 17A in Helicobacter pylori-Infected patients with gastric premalignant lesions.

In my opinion the content of the manuscript, its aim and the direction are clear. The manuscript is original and its topic is interesting. The title expresses clearly the content of the manuscript and highlights the importance of the study. The introduction section clearly summarize the current state of the topic as well as clearly define the aim of the study. I suggest using only one terminology -  H. pylori or  Helicobacter pylori

Study design and methods are appropriate for the research question. The results are presented clearly and accurately and are consisted with the aim of the work and the methods. All the relevant data have been included in the article.

The authors logically explain and describe their findings. The limitations of the study werw not mentioned, I suggest that the authors present the possible limitations/difficulties in this study.

The authors concluded that serum determination of interleukin-17A would be helpful in the clinical practice of H. pylori-infected patients, and eventually for predicting the development of gastric cancer. I suggest that conclusions should be more detailed, as presented in abstract -    the results obtained indicate that H. pylori is able to drive gastric IL-17 inflammation in IM/DYS H. pylori-infected patients, and that IL-17A serum levels are significantly increased in H. pylori-infected patients with IM/DYS. In addition, in this chapter the rows 293/294 is repeated.

I suggest a revision of the References, because of the 45 citations, a number of 14 citations (representing 31%) are from other articles published before the year 2000.

Author Response

- I suggest using only one terminology -  H. pylori or Helicobacter pylori

Helicobacter pylori” has been used throughout the text.

- The limitations of the study wer not mentioned, I suggest that the authors present the possible limitations/difficulties in this study.

In order to address this point we introduced the following text to the revised versione of the manuscript: “Given that this investigation represents a pilot single-center study, future multi-center research activities dealing with serum IL-17A levels in gastric adenocarcinoma would be very important for a better definition of the test.” (line 306)

The authors concluded that serum determination of interleukin-17A would be helpful in the clinical practice of H. pylori-infected patients, and eventually for predicting the development of gastric cancer. I suggest that conclusions should be more detailed, as presented in abstract - the results obtained indicate that H. pylori is able to drive gastric IL-17 inflammation in IM/DYS H. pylori-infected patients, and that IL-17A serum levels are significantly increased in H. pylori-infected patients with IM/DYS. In addition, in this chapter the rows 293/294 is repeated.

The conclusions have been more detailed in the revised version of the manuscript (line 316).

The repetition has been deleted.

I suggest a revision of the References, because of the 45 citations, a number of 14 citations (representing 31%) are from other articles published before the year 2000.

References have been increased with updated articles and reviews, both in the Introduction and in the Discussion. The new references have been highlighted into the References paragraph.

Round 2

Reviewer 1 Report

I thank the authors for their response to my suggestions on the initial version of the manuscript and I congratulate them for this excellent work.